# Role of BamHI-A Rightward Frame 1 in Epstein–Barr Virus-Associated Epithelial Malignancies

**DOI:** 10.3390/biology9120461

**Published:** 2020-12-11

**Authors:** Rancés Blanco, Francisco Aguayo

**Affiliations:** 1Programa de Virología, Instituto de Ciencias Biomédicas (ICBM), Faculty of Medicine, Universidad de Chile, Santiago 8380000, Chile; rancesblanco1976@gmail.com; 2Universidad de Tarapacá, Arica 1000000, Chile; 3Advanced Center for Chronic Diseases (ACCDiS), Faculty of Medicine, Universidad de Chile, Santiago 8380000, Chile

**Keywords:** Epstein–Barr virus, epithelial carcinogenesis, BARF1

## Abstract

**Simple Summary:**

Epstein–Barr virus is a ubiquitous persistent virus, which is involved in the development of some human cancers. A licensed vaccine to prevent Epstein–Barr virus infection is lacking. BamHI-A rightward frame 1 is a viral protein specifically detected in both nasopharyngeal and Epstein–Barr virus-positive gastric cancers. It has been proposed that this viral protein confers cancer properties to infected epithelial cells and is involved in the escape of cancer cells from immune recognition. In this review, we summarize the properties of BamHI-A rightward frame 1 which confers cancer characteristics to infected epithelial cells. Thus, BamHI-A rightward frame 1 is a potential therapeutic target for the treatment of either Epstein–Barr virus (EBV)-positive nasopharyngeal or gastric cancers.

**Abstract:**

Epstein–Barr virus (EBV) infection is associated with a subset of both lymphoid and epithelial malignancies. During the EBV latency program, some viral products involved in the malignant transformation of infected cells are expressed. Among them, the BamHI-A rightward frame 1 (BARF1) is consistently detected in nasopharyngeal carcinomas (NPC) and EBV-associated gastric carcinomas (EBVaGCs) but is practically undetectable in B-cells and lymphomas. Although BARF1 is an early lytic gene, it is expressed during epithelial EBV latency, mainly as a secreted protein (sBARF1). The capacity of sBARF1 to disrupt both innate and adaptive host antiviral immune responses contributes to the immune escape of infected cells. Additionally, BARF1 increases cell proliferation, shows anti-apoptotic effects, and promotes an increased hTERT activity and tumor formation in nude mice cooperating with other host proteins such as c-Myc and H-ras. These facts allow for the consideration of BARF1 as a key protein for promoting EBV-associated epithelial tumors. In this review, we focus on structural and functional aspects of BARF1, such as mechanisms involved in epithelial carcinogenesis and its capacity to modulate the host immune response.

## 1. Introduction

The human gammaherpesvirus-4 (HHV-4), commonly referred to as Epstein–Barr virus (EBV), is a member of the Herpesviridae family and *Lymphocryptovirus* genus [1]. EBV establishes a latent persistent infection affecting more than 90% of the human population worldwide [2]. Primary EBV infection in children usually occurs without any symptoms. Conversely, during adolescence and early adulthood primary EBV infection may produce infectious mononucleosis (IM) disease, which is characterized by an IgM antibody response against EBV, the circulation of increased loads of latently infected B-cells, and the development of EBV-specific CD8 + T cells [3]. The circulating CD8 + T cells recognizing lytic EBV antigens are detected approximately five days after the appearance of IM symptoms and are responsible for the specific immune response against EBV-infected cells [4,5].

In 2018, an estimated 200,000 newly diagnosed cancers were related to EBV infection [2,6], including both lymphoid and epithelial malignancies. According to the International Agency for Research on Cancer (IARC), only three epithelial tumors (nasopharyngeal, gastric, and lymphoepithelial carcinomas) have proved to be undoubtedly associated with EBV infection [7,8,9]. On the other hand, this virus has been found in tumors of the oral cavity, breast, and uterine cervix, among others, which indicates the need for further investigation. Among the EBV proteins involved in the malignant transformation of epithelial cells, the BamHI-A rightward frame 1 (BARF1) is of utmost importance [10]. This lytic gene is highly expressed in nasopharyngeal carcinomas (NPC) and EBV-associated gastric (EBVaGC) carcinomas during latency [11,12], but is virtually undetectable in B-cells and lymphomas, in which it can mostly be found during the viral lytic cycle [13,14]. This fact allows for the consideration of BARF1 as an epithelial-specific EBV oncogene as well as an attractive potential therapeutic target for EBV-associated epithelial tumors [15]. Previously, the therapeutic potential of BARF1 has been extensively reviewed [16]. In this review, the structure and biological functions of BARF1 which explain its role in cancer have been summarized. Finally, a model of BARF1 mediated carcinogenesis in epithelial cells is proposed. 

## 2. EBV Structure and Replication

The EBV genome is a linear double-stranded DNA of approximately 172 kilobase pairs in length, enclosed by a nucleocapsid formed by 162 capsomers. Tegument proteins fill the space between the inner icosahedral capsid and the envelope which contains virus-encoded glycoproteins that form surface spikes [17,18]. EBV primarily infects two different cell types, B-cells and epithelial cells. EBV entry in B-cells is mediated by an interaction between the viral glycoprotein gp350 and either CD21 or CD35 lymphocyte receptors. This interaction also involves EBV gp42 binding with B-cell MHC-II [2,19]. In epithelial cells, the membrane fusion is initiated by an interaction of the viral glycoprotein H (gH) and glycoprotein L (gL) heterodimer (referred to as gH/gL) and αvβ6, αvβ8, or αvβ5 cell surface integrins [20,21]. Besides, EBV expresses the BMRF2 glycoprotein, which binds to α3, α5, and β1 integrins promoting infection of polarized epithelial cells through their basolateral membrane domain [22]. Moreover, it was also reported that ephrin receptor tyrosine kinase A2 (EphA2) mediates the EBV entry into epithelial cells [23]. After EBV binding to cell receptors, the viral envelope and cell membrane are fused, the viral genome traffics to the nucleus where the virus establishes latent or lytic phases of infection. EBV can establish a long-term latency with eventual lytic reactivations in B-cells, while EBV only establishes lytic infections in normal epithelial cells. In fact, latent forms of EBV are almost not detected in normal nasopharyngeal or oral epithelial cells where EBV only establishes a lytic cycle [24]. It has been reported that latently infected epithelial cells are detected in tonsil explants in the presence of acyclovir, although in less than 0.01% of cells [25]. The EBV replicative cycle in epithelial cells is less understood, at least in part due to the historical absence of an in vitro model for efficient viral replication. However, a model for EBV replication based on organotypic epithelial cell cultures was described, which demonstrated that EBV replicates in primary stratified epithelium without cells exclusively expressing latency genes [26,27,28].

The switch from latency to lytic cycle involves the expression of BZLF1 and BRLF1 genes, which encode for the immediately early (IE) Zta and Rta proteins, respectively [29,30]. Both Zta and Rta proteins are transcription factors that regulate EBV early lytic cascade, being actively transcribed previously to lytic viral replication. Zta and Rta expression is regulated by the Zp and Rp promoters. It has been suggested that Blimp1, expressed during terminal differentiation of epithelial cells, is important for Zp promoter activation allowing lytic cycle activation [31]. Additionally, it has been described that some chemicals such as 12-O-tetradecanoylphorbol-13-acetate (TPA), sodium butyrate and calcium ionophores induce the EBV lytic cycle. Epigenetic modifications such as DNA methylation and histone deacetylation are related to inhibition of IE gene transcription [32]. Anyhow, the expression of both Zta and Rta proteins is ever required for subsequent expression of lytic early genes [33]. There are some early promoters regulated by these IE proteins such as BMRF1, SM, BHLF1, and BHRF1. The genes that are currently translated at this stage are the viral DNA polymerase (BALF5) [34], DNA polymerase processivity factor (BMRF1) [35], helicase (BBLF4) [36], primase (BSLF1) [36], and others. BMRF1 and BRRF1 are transcription factors that activate the oriLyt (lytic replication origin). This oriLyt has a complex structure that contains multiple regions required for DNA replication which is achieved by the viral BALF5 DNA polymerase [37]. The viral DNA replication occurs through a rolling circle-mechanism conducting to the formation of concatemers which are finally cleaved and packaged [38,39]. Once the viral DNA is replicated, late lytic genes are expressed, but little is known about how EBV late promoters are regulated. The late genes encode for structural proteins including nucleocapsid and glycoproteins of the viral envelopment (gp350/220, gp85, gp42, and gp25). In epithelial cells, late gene expression and viral maturation (lytic cycle) occur in the upper differentiated layer of stratified epithelia [28]. In the lytic phase, all of the EBV products are expressed with virions being assembled and released. Then, viral progeny can display cell-to-cell spread to infect new hosts [40]. In latency, the EBV genome persists in an episomal form in the nucleus of memory B-cells with a restricted production of viral proteins and transcripts [41,42]. This silent mode of infection reduces the potential host immune response against viral proteins [43]. Specifically, the Epstein-Barr-nuclear antigens (EBNA) 1, 2, 3A-C, and LP; the latent membrane proteins (LMP) 1 and 2A-B as well as the non-polyadenylated Epstein-Barr-encoded small RNAs (EBER) 1 and 2 can be expressed during EBV latent program (reviewed in [2,44,45]). Additionally, approximately 44 different mature miRNAs are expressed from an EBV genome region located in two opposite regions named BamHI fragment H rightward open reading frame 1 (BHRF1) and BamHI A rightward transcripts (BART) (Reviewed in: [44,45]).

## 3. Structure of BARF1 Protein

BARF1 gene encodes a 221 amino acid protein which is structured in two domains and is included in the immunoglobulin fold family [46,47,48]. The N-terminal domain ranges from residues 21 to 123 and the C-terminal domain from residues 125 to 220 [46] (Figure 1). BARF1 is cleaved after the first 20 amino acids and mostly secreted by EBV-infected epithelial cells as a soluble hexameric molecule (sBARF1). The hexameric rings are formed by three head-to-tail dimers of the BARF1 protein arranged in two layers [46]. In culture media from NPC-derived BARF1-expressing HEK-293 cells, BARF1 monomer was detected as a 27–29 kDa band by Western blot analysis under reducing conditions. Also, non-reducing conditions revealed the hexameric form of BARF1 ranges between 160 and 180 kDa [48]. BARF1 is synthesized in the endoplasmic reticulum and post-translationally modified in the Golgi complex with a high mannose (GlcNac2-Man9) N-linked glycosylation on the asparagine 95 (Asn95) residue. N-glycosylation of BARF1 plays a crucial role in folding, subcellular translocation, and final secretion [47]. Besides, BARF1 contains an O-linked glycosylation site located at threonine 169 (Thr169) and represented by a trisaccharide sugar structure [46]. It is also phosphorylated on serine and threonine residues [49,50]. The sequence of BARF1 is highly conserved, although some variations patterns have been reported. In northern Chinese samples, the BARF1 gene sequencing revealed 13 amino acid mutations, among them V29A, V46A, D79G, V113I, and D138Y, which were the most frequent. Interestingly, the V29A mutation was mostly evidenced in NPC samples (25.3%, 20/79) compared to EBVaGC cases (0/45) or healthy donors (4.3%, 2/46) [51]. NPC cases from northern China showed a higher frequency of V29A variant than NPC cases from southern China. A similar result was obtained when the groups of healthy donors were compared [52]. In 80.3% of NPC samples from Indonesian patients, 3 main substitutions (V29A, W72G, H130R) in the BARF1 gene sequence were evidenced; these were considered unable to alter the tertiary structure or function of this protein [53]. In fact, none of these sequence variations are located in relevant functional domains of BARF1 [50].

It was reported that stable transfection of HEK-293 or gastric cancer (GC) cells with BARF1 encoding vector allows secretion of this protein to the culture medium [10,48]. The transient expression of BARF1 in NPC and GC cells also leads to the same results [48]. In contrast, perinuclear localization of BARF1 was observed in BARF1-Flag-transfected HaCaT cells, while in NPC, this protein was evidenced in both cytoplasm and plasma membrane [54,55]. Moreover, BARF1 was detected on the cell surface of NPC and GC cells [15]. BARF1 is secreted via the endoplasmic reticulum-Golgi apparatus (ER-GA) classical pathway [48], which supports the cytoplasmic and membrane localization. Interestingly, the cellular uptake of purified secreted BARF1 (sBARF1) and subsequent nuclear localization was evidenced in human keratinocytes although a weak amount of BARF1 was found in the nucleus [55]. The trafficking of sBARF1 from extracellular media to the cell nucleus could be a mechanism by which this protein exerts some intracellular functions. However, the absence of BARF1 nuclear localization was reported in NPCs [54].

## 4. Expression of BARF1 in EBV-Related Epithelial Tumors

During epithelial latency, some early lytic genes, but rarely late genes, are detected which contribute to abortive lytic infection [56,57,58]. The potential contribution of lytic genes to viral tumorigenesis has previously been reviewed [59]. Besides, antibody levels for lytic proteins were associated with TNM (tumor, node, metastasis) stages in NPCs, supporting the role of abortive lytic infection in the progression of epithelial malignancies [60]. BARF1 is an early lytic gene and its expression is activated by the immediate early proteins Zebra and Rta [61]. However, in latency, this lytic gene is consistently expressed in EBV-associated epithelial tumors but not in lymphomas, which allows the consideration of BARF1 as a carcinoma-specific EBV protein [11,12]. In NPC, both BARF1 and LMP1 proteins are commonly detected. However, in EBVaGC BARF1 is detected in the absence of LMP1, suggesting that BARF1 is the most important EBV oncogene in these malignant tumors [50].

BARF1 mRNA was detected in 74.4% of nasopharyngeal brushings from NPC patients [62] as well as in 69.2%–87% of NPCs [56,63,64]. In these studies, the expression of some latent proteins such as EBNA1, LMP1, and LMP2 was also detected, although the presence of lytic proteins was not assessed. Seto et al. reported BARF1 expression in 93.4% and 83.3% of EBV-positive NPC and GC samples, respectively, in the absence of some lytic genes such as BZLF1, BMRF1, and BLLF1 [65]. In another study, BARF1 and BZLF1 were detected in 75.0% and 77.5% of GCs, respectively, while BcLF1 (a late EBV protein) levels were slightly decreased (62.5%) [66]. Wang et al. detected BARF1/BZLF1 and BHRF1 transcripts in 46% and 15.4% of EBVaGCs, respectively [63]. Similar results were described by Lu et al. [64]. Furthermore, BcLF1 mRNA was found in 63.6% of GCs, but not when other lytic genes (BRLF1 and BLLF1) were analyzed. In EBVaGC, the expression of BARF1 was evidenced along with EBNA1 and LMP2A latent genes, reaching 100% and 36.4% of positivity, respectively [64].

Interestingly, it has been reported that the BARF1 promoter region remains highly methylated in both epithelial and B-cells [61]. A similar result was found in EBVaGC tissues [67]. Conversely, BARF1 was expressed regardless of the promoter methylation status in NPC cells as well as in the latently EBV-infected C15 and C17 NPC xenografts. In line with this, transfection of NPC cells with a vector expressing the Rta protein was able to increase BARF1 mRNA levels. These facts suggest that BARF1 transcription can revert the silencing induced by its promoter methylation by Rta protein activation [61]. Additionally, the capacity of ΔNp63α isoform to specifically transactivate the BARF1 promoter was evidenced in GC cells [68]. ΔNp63α is a factor of epithelial cell differentiation, which is found to be overexpressed in both NPC and GC [51,69]. Interestingly, BARF1 promoter transactivation induced by ΔNp63α was irrespective of its methylation status, shedding light on a potential mechanism by which BARF1 protein is expressed in epithelial cells during EBV latent infection [68]. However, evidence concerning BARF1 protein expression in NPC and GC is scarce, probably due to the rapid secretion of this protein to the extracellular medium. BARF1 protein was detected in 2/7 (28.6%) [65] and 23/27 (85.2%) [54] of NPC biopsies. Conversely, other studies failed to demonstrate BARF1 protein in NPC or GC tissues, which in turn were positive for BARF1 mRNA expression [50]. Additionally, it was reported that BARF1 transcripts are almost exclusively expressed in the nucleus of SNU719 GC cells [70]. In this respect, further studies addressing BARF1 protein expression in EBV-positive epithelial tumors are warranted.

## 5. BARF1 and Cell Proliferation Rates

BARF1 interacts with some cell cycle-regulating proteins, promoting epithelial cell proliferation. In fact, the mitogenic activity of BARF1 was demonstrated in primary epithelial cells treated with sBARF1 which showed increased cell proliferation when compared to control cells [71]. An increased proliferation rate was also evidenced in BARF1-transfected HaCaT cells compared with the mock control. Moreover, exogenous BARF1 treatment of HaCaT cells increased its transition from G1 to S-phase [55]. Nevertheless, no significant differences in the percentage of S-phase cells were evidenced between BARF1-transfected GC cells and controls [47]. In HaCaT cells, BARF1 was demonstrated to increase the expression of cyclin D1 at transcriptional and protein levels [72]. After activation, cyclin D1 forms complexes with cyclin-dependent kinases (CDK) 4 and 6 mediating the progression from G1 to S-phase, whereas the tumor suppressor p21^WAF1^ can arrest the cell cycle transition by inhibiting CDKs [73,74]. Additionally, increased cell proliferation was evidenced in EBV-negative cells stably transfected with a BARF1 encoding vector [10]. Notably, in GC cells, transfection with the BARF1 gene promoted a reduction in p21^WAF1^ expression [75], suppressing one of the most important regulatory mechanisms of cell proliferation. The main effects of BARF1 in cell cycle deregulation are summarized in Table 1.

Increased proliferation rates were associated with activation of nuclear factor kappa B (NF-κB) RelA/cyclin D1 signaling pathway in GC cells expressing BARF1 [75]. Furthermore, the phosphorylated form of NF-κB inhibitor IκBα was increased in these transfected cells, promoting NF-κB nuclear translocation to initiate transcription of κB-dependent genes [77]. BARF1 also promotes cell proliferation by increasing NF-κB RelA and upregulating the microRNA-146a-5p, which in turn downregulates SMAD4 [10]. The inactivation of SMAD4, a critical mediator of the growth-inhibiting TGFβ signaling pathway, reduces the expression of some CDK inhibitors (e.g., p15, p21, and p27), resulting in uncontrolled cell proliferation [78]. Also, reduced cell proliferation was observed when EBV+ GC cells were transfected with a small interfering RNA (siRNA) for BARF1 knockdown [10].

## 6. Anti-Apoptotic Effects of BARF1

BARF1 protects epithelial cells from the intrinsic cell death pathway by regulating anti-apoptotic (e.g., Bcl-2, Bcl-xL) and pro-apoptotic (e.g., Bax) pathways. In fact, it was previously reported that the N-terminal region of BARF1 protein (codons 1 to 54) is responsible for activation of the anti-apoptotic Bcl-2 [79]. Transfection of primary epithelial and NPC cells with the BARF1 gene induces increased Bcl-2 levels [12]. Similarly, Bcl-xL upregulation was evidenced in HaCaT BARF1-transfected cells when compared with BARF1-negative control cells [72]. Furthermore, BARF1-mediated Bcl-2 and Bcl-xL upregulation was associated with activation of c-Jun N-terminal protein kinase (JNK) 1/2/3, p38 mitogen-activated protein kinase (MAPK), and extracellular signal-regulated protein kinases 1 and 2 (ERK1/2)/c-Jun cascades in GC cells [80]. Another study reported the capacity of BARF1 to protect GC cells from apoptosis by increasing the Bcl-2/Bax ratio [47]. Pro-apoptotic Bax is directly activated by p53 in response to DNA damage and acts to neutralize the anti-apoptotic effect of Bcl-2. In fact, this protein can depolarize the mitochondria, inducing Cytochrome-C release, which then activates the caspase cascade (reviewed in [81]). Moreover, a transcriptomic approach showed caspases downregulation, including caspase 3, in GC cells ectopically expressing BARF1 [47]. The increase in the Bcl-2/Bax ratio was similarly evidenced in GC cells expressing BARF1 after Taxol (paclitaxel) exposure. Likewise, a significant reduction in the percentage of these cells showing late apoptosis events (nuclear fragmentation) was evidenced in the same conditions [47], suggesting a potential contribution of BARF1 to apoptosis-based therapy resistance in EBVaGC.

The anti-apoptotic effects of BARF1 were also evaluated in NPC cells transfected with recombinant EBV (rEBV) carrying the BARF1 gene (BARF1-rEBV) [82]. Interestingly, an increase in the resistance to apoptosis was evidenced in BARF1-rEBV-infected NPC cells measured as nuclear fragmentation upon serum depletion. In contrast, no anti-apoptotic effect of BARF1 was detected in CNE-1 cells using the same system. Additionally, no changes in the percentage of cells showing early apoptotic events were reported when GC cells expressing BARF1 were analyzed [75]. This fact suggests that BARF1 exerts its anti-apoptotic effects in a cell type-dependent manner, although further studies are necessary to elucidate these controversial results.

## 7. Immortalization and Tumorigenic Properties of BARF1

Telomere elongation by the telomerase enzyme is a prerequisite by which cells can reach unlimited replicative potential and also contributes to tumorigenic properties [83]. Increased telomerase activity was reported in BARF1-transfected epithelial cells, which was comparable to that obtained in human telomerase reverse transcriptase (hTERT)-transfected cells, allowing these cells to escape from senescence [12]. In the same study, it was demonstrated that hTERT activation in BARF1-transfected cells is accompanied by c-Myc upregulation [12]. It is known that c-Myc is an important transcriptional regulator of hTERT, which can directly increase its expression through its interaction with binding motifs located in the TERT promoter [84]. This fact suggests a potential synergism between BARF1 and c-Myc to induce hTERT activation, leading to epithelial cell immortalization. Moreover, BARF1 directly binds to initiator elements located at positions +13 and +43 in the hTERT promoter, which induces telomerase expression in epithelial cells [12]. The potential involvement of BARF1 in the protection of telomeres to prevent their shortening was also suggested [85], although further studies are needed to prove this hypothesis. On the other hand, BARF1 was able to induce anchorage-independent growth in soft agar as well as altered migration of HEK-293 cells [50]. Furthermore, the infection of NPC cells with EBV carrying the BARF1 gene induced tumor growth in nude mice, but not in EBV-infected cells lacking BARF1 [82]. However, in other NPC cells (NP69) the transfection with BARF1 only provided proliferative advantages and increased anchorage-independent growth in cooperation with H-ras [12].

In fact, normal nasopharyngeal cells coexpressing BARF1 and H-ras were capable of inducing tumor formation in nude mice but this effect was not observed in cells expressing BARF1 or H-ras alone. Similarly, BARF1 expression in primary epithelial cells was insufficient to induce tumor formation in nude mice [86]. Taken together these data suggest that BARF1 tumorigenic properties depend on its cooperation with other oncogenes. BARF1 expression was detected in xenografts from NPC and GC cells which were grown in nude mice [54,65]. Interestingly, EBNA1 was undetectable in NPC xenografts [87], although this viral product was detected in GC xenografts [88]. Altogether, these results suggest a central role for BARF1 in the tumorigenicity of NPC and GC cells in vivo, although other factors are required for malignant transformation.

## 8. BARF1 Expression and Modulation of Host Immune Response

BARF1 also contributes indirectly to epithelial carcinogenesis by promoting evasion of both innate and adaptive immune responses. This viral protein is responsible for the sequestration of the macrophage colony-stimulating factor (M-CSF, also known as CSF-1), inducing a disruption in the differentiation and activity of macrophages [89,90]. For instance, the hijack of M-CSF by sBARF1 induces a reduction in the expression of a variety of macrophage differentiation-specific markers such as CD14, CD11b, CD16 and CD169 [61]. This fact also interferes with the function of mononuclear cells, by inhibition of interferon-alpha (IFN-α) production and release [76]. IFN-α is an early cytokine that plays an important role in the host anti-viral immune response. Moreover, M-CSF pre-incubation with sBARF1 inhibited M-CSF receptor, Akt, and MAPK phosphorylations in myeloid leukemia cells, which attributes a role of BARF1 in the survival and proliferation capacity of macrophages [61]. A close relationship between BARF1 structure and CD80 (costimulatory molecule expressed by antigen-presenting cells) was also reported. This homology could allow sBARF1 to interfere with T-cell activation mediated by the co-stimulatory effect of CD80, which is expressed by pro-inflammatory macrophages (M1) [50]. A low density of M1 tumor-associated macrophages correlated with a decreased oxidative stress (OS) of GC patients [91], while low expression of CD80 in NPC was also associated with poor survival [75]. Taken together, evasion of macrophage-mediated innate antitumor response constitutes a central role of sBARF1.

The gene expression profile of HaCaT BARF1-transfected cells revealed the downregulation of some human cytokines also related to the host immune response such as human interleukin 1-alpha (IL-1α) and interleukin 8 C-terminal variant (IL-8) [72]. IL-1α is a pro-inflammatory cytokine usually produced by cells of the immune system but also by epithelial cells, including NPC [92]. Although, the functions of IL-1α are diverse, when this cytokine is expressed in the cytoplasmatic membrane it can induce an anti-tumor immune response [93]. Similarly, IL-8 can be released by epithelial cells, resulting in polymorphonuclear neutrophils and other immune cells recruitment to the infection site [94]. Although the capacity of CD4+ and CD8+ lymphocytes to specifically react to BARF1 protein in NPC patients was found, this immune response was 5–8-fold lower compared with those generated for EBNA1, other immunodominant EBV protein [95,96]. Moreover, humoral immune response against BARF1 (IgA and IgG) found in NPC patients was lower when compared with other EBV-related proteins and also similar to the antibody response obtained in healthy EBV-seropositive persons [97,98]. This lower immunogenicity of BARF1 could also contribute to tumor escape from the host immune system. Figure 2 summarizes some mechanisms by which BARF1 contributes to epithelial cells carcinogenesis and immune evasion.

## 9. Conclusions 

During EBV abortive lytic infection, both latent and lytic genes are expressed to contribute to viral carcinogenesis. BARF1 is an early lytic viral protein highly expressed in latently EBV-infected epithelial cells, but it is less frequently detected in lymphomas. BARF1 protein is almost completely secreted by EBV-infected epithelial cells, although intracellular BARF1 confers increased proliferation rates, apoptosis protection, and tumor properties, synergizing with other oncogenes such as H-ras. The fact that BARF1 is expressed in the absence of LMP1 in EBVaGC could indicate that this protein plays a central role in the carcinogenesis of EBV-infected epithelial cells. Nonetheless, the function of BARF1 in other epithelial malignancies where EBV infection has been detected needs further investigation. Based on the structure and role of BARF1, immunotherapeutic approaches raised against this protein could impact the biological behavior of EBV-associated epithelial tumors.

## Figures and Tables

**Figure 1 biology-09-00461-f001:**
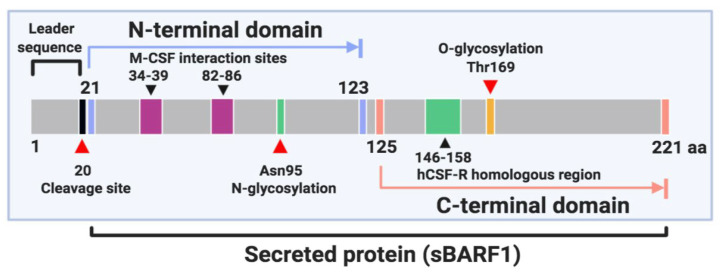
Schematic representation of BamHI-A rightward frame 1 (BARF1) protein.

**Figure 2 biology-09-00461-f002:**
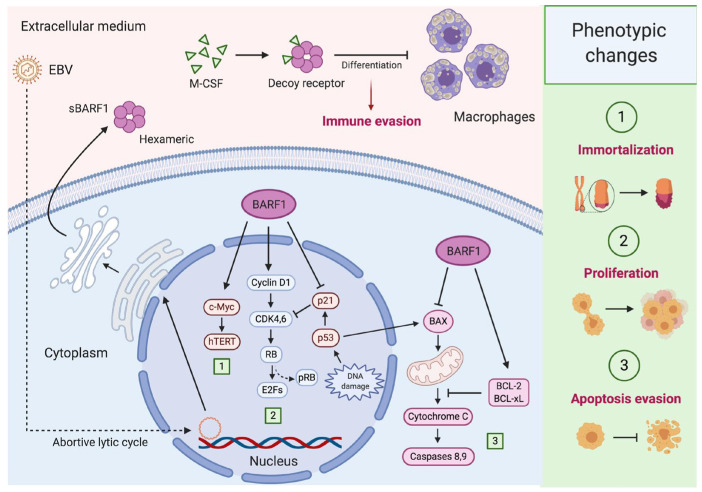
BARF1-mediated carcinogenesis in epithelial cells. During epithelial Epstein–Barr virus (EBV) latency (I/II), some lytic genes such as BARF1 are expressed (abortive lytic infection), cooperating with latent genes for carcinogenesis. (1) BARF1 synergizes with c-Myc to promote hTERT activation, avoiding replicative senescence and leading to epithelial cell immortalization; (2) BARF1 also increases cyclin D1 levels promoting the progression from G1 to S-phase of the cell cycle. Moreover, BARF1 induces a reduction in p21^WAF1^ levels disrupting one of the most important regulatory mechanisms of cell proliferation; (3) BARF1 rescues epithelial cells from apoptosis reducing the expression of the pro-apoptotic Bax and increasing the anti-apoptotic proteins Bcl-2 and Bcl-xL. Furthermore, BARF1 is secreted as a hexameric protein (sBARF1) which contributes to immune evasion. In fact, sBARF1 sequestrates the macrophage colony-stimulating factor (M-CSF), inducing a disruption in the differentiation and activation of these cells. Created by BioRender.com.

**Table 1 biology-09-00461-t001:** Contribution of BARF1 expression to epithelial cell carcinogenesis and impaired host immune response.

Target Molecule/ Pathway	Biological Effect	Cell Line and Origin	Ref.
	BARF1 Increases Cell Proliferation Rates		
Cyclin D1	Increases cyclin D1 on both transcriptional and protein level	HaCaT immortalized human keratinocytes	[72]
p21^WAF1^	Reduce the expression of the tumor suppressor p21WAF1	SNU601 GC cells	[75]
NFκB RelA	Increases NF-κB RelA upregulating the microRNA-146a-5p and downregulating SMAD4 Increases NF-kB RelA/cyclin D1 signaling augmenting the phosphorylated form of NF-kB inhibitor IkBα	SNU601 GC cells	[10] [75]
	BARF1 protects cells from apoptosis		
Bcl-2	Increases the expression of the anti-apoptotic Bcl-2	NP69 NPC cells	[12]
Bcl-xL	Increases the expression of the anti-apoptotic Bcl-xL	HaCaT cells	[72]
Bcl-2/Bax	Increases the Bcl-2 to Bax ratio	BGC283 GC cells	[47]
Caspase 3	Downregulates the executor of apoptosis caspase 3	BGC283 GC cells	[47]
	BARF1 induces cell immortalization		
hTERT	Increases telomerase activity in cooperation with c-Myc Increase telomerase activity binding to initiator elements at positions +13 and +43 in hTERT promoter region	PATAS primary monkey kidney epithelial cells PATAS and HeLa cervical cells	[12] [12]
	BARF1 modulates the host immune response		
M-CSF	Sequesters M-CSF reducing the macrophage differentiation markers CD14, CD11b, CD16 and CD169 Sequesters M-CSF reducing the production of IFN-α Sequesters M-CSF inhibiting the phosphorylation of M-CSF receptor, Akt and MAPK	PBMC isolated monocytes-derived macrophages Adherent human mononuclear cells from PBMC M-CSF-dependent MUTZ-3 cells	[61] [76] [61]

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
