# Peer review of "Role of BamHI-A Rightward Frame 1 in Epstein–Barr Virus-Associated Epithelial Malignancies"

_biology, 2020, doi:10.3390/biology9120461_

Round 1
Reviewer 1 Report
The authors reviewed the expression of BARF1 and its effects on proliferation, apoptosis, and immune response in EBV-associated epithelial tumors, and indicated that BARF1 expresses as a latent gene and oncogene in epithelial tumors, might be a very promising therapeutic target. It is worth noting that a very similar review was published on Cancers (Basel) (PMID: 32708965; PMCID: PMC7409022; DOI: 10.3390/cancers12071940), and the author should highlight the key points.
In addition, the author needs to pay attention to the italics of BARF1. There are several italics in the text, but it is not unified with the previous text.
Line 167, Seto et al. reported BARF1 expression in 93.4% and 83.3% of NPC and GC samples, respectively, …… Should it be in 93.4% and 83.3% of EBV-positive NPC and GC samples?
The structure of Table 1 should be reorganized so that the same genes are placed in the same column
Author Response
Comment 1: The authors reviewed the expression of BARF1 and its effects on proliferation, apoptosis, and immune response in EBV-associated epithelial tumors, and indicated that BARF1 expresses as a latent gene and oncogene in epithelial tumors, might be a very promising therapeutic target. It is worth noting that a very similar review was published on Cancers (Basel) (PMID: 32708965; PMCID: PMC7409022; DOI: 10.3390/cancers12071940), and the author should highlight the key points.
Answer 1: Thanks for this comment. The review by Kwok-Fung Lo et al (2020) focus on therapeutic potential of BARF1. Additionally, the authors described immunomodulatory and some oncogenic properties of this viral protein. In this new review, we focus on mechanisms and biological functions of BARF1 which can explain its role in epithelial cancer. Thus, we think that both reviews are complementary. A sentence was added in the Introduction.
Comment 2: In addition, the author needs to pay attention to the italics of BARF1. There are several italics in the text, but it is not unified with the previous text.
Answer 2: Thanks for this comment. The manuscript was carefully reviewed, and the use of italics was eliminated (lines 262, 266, 275, 282).
Comment 3: Line 167, Seto et al. reported BARF1 expression in 93.4% and 83.3% of NPC and GC samples, respectively, …… Should it be in 93.4% and 83.3% of EBV-positive NPC and GC samples?
Answer 3: Thanks for this comment. This sentence was corrected (Line 169).
Comment 4: The structure of Table 1 should be reorganized so that the same genes are placed in the same column.
Answer 4: Thanks for this comment. Table 1 was reorganized (line 215).
Reviewer 2 Report
In the review article "Role of BamHI-A rightward frame 1 in Epstein-Barr virus-associated epithelial malignancies" by Rancés Blanco and Francisco Aguayo, the authors evaluate recent knowledge of the physiological functions of EBV BamHI-A rightward frame 1 (BARF1) in the specific contest of Nasopharyngeal Carcinoma (NPC) and Gastric Carcinoma (GC) tumor microenvironment.
The well-written discussion presented in the review is clear and to the point: it provides valuable information about the pathways themselves and the ability of BARF1 to modulate various aspects of viral carcinogenesis in EBV-infected epithelial cells. It is a pleasure to read the article: the authors presented a serious comprehensive analysis with well-described original material and references. Each section provides efficient information on all necessary aspects of the subject.
Many recent studies indicate that oncogenic viruses are significant for the progression of infection-associated cancers. Although the involvement of EBV latency gene products in the development and progression of epithelial malignancies is well established, the contribution of abortive lytic cycle and specifically BARF1 to EBV pathogenesis in NPC and GC has only recently begun to be elucidated. The clear significance of this analytical study will attract broad general interest. This review will be interesting to many researchers in the areas of oncology associated with tumor viruses.
Author Response
Thank you very much for these comments.
Reviewer 3 Report
In this review, the authors describe the role of BRAF1 in Epstein-Barr virus (EBV) -positive epithelial cells. It's a very unique review, but it lacks some essential information and needs to be added.
- Many groups have reported that expression of BARF1 protein is observed in nasopharyngeal cancer (Seto E et al. J Med Virol, 2005). Although expression of BRAF1 RNA has been confirmed in EBV-related gastric cancer (EBVaGC), but there is no report that protein has been detected yet. This needs to be noted in the review. Non-coding RNAs, such as circular RNA, can play an important role, even in the absence of protein expression. For example, it has been reported that BARF1 RNA expression is found only in the nucleus in EBVaGC (Jang BG et al. Cancer Res Treat, 2011).
- M-CSF is also known as CSF-1, and many EBV papers also describe it as CSF-1. You need to write this description once. In addition, Strockbine LD et al. J Virol, 1998 was the first to report that BARF1 functions as a CSF-1 soluble receptor, and this paper must be cited in this review.
- BARF1 was originally identified as an early gene for lytic infection, but is a unique gene that is also expressed during latent infection. It needs to be emphasized that it is a unique gene.
4) R should change to Rta.
Author Response
In this review, the authors describe the role of BRAF1 in Epstein-Barr virus (EBV) -positive epithelial cells. It's a very unique review, but it lacks some essential information and needs to be added.
Comment 1: Many groups have reported that expression of BARF1 protein is observed in nasopharyngeal cancer (Seto E et al. J Med Virol, 2005). Although expression of BRAF1 RNA has been confirmed in EBV-related gastric cancer (EBVaGC), but there is no report that protein has been detected yet. This needs to be noted in the review. Non-coding RNAs, such as circular RNA, can play an important role, even in the absence of protein expression. For example, it has been reported that BARF1 RNA expression is found only in the nucleus in EBVaGC (Jang BG et al. Cancer Res Treat, 2011).
Answer 1: Thank you very much for this comment. The limited data concerning BARF1 expression at protein level was included in the manuscript. It was also included a commentary concerning the preferential expression of BARF1 RNA in the nucleus of SNU719 GC cells (Lines 189-196).
Comment 2: M-CSF is also known as CSF-1, and many EBV papers also describe it as CSF-1. You need to write this description once. In addition, Strockbine LD et al. J Virol, 1998 was the first to report that BARF1 functions as a CSF-1 soluble receptor, and this paper must be cited in this review.
Answer 2: Thank you very much for this comment. The description of CSF-1 was included in the manuscript (line 287).The report of Strockbine LD et al. 1998 was cited in the manuscript (line 288).
Comment 3: BARF1 was originally identified as an early gene for lytic infection, but is a unique gene that is also expressed during latent infection. It needs to be emphasized that it is a unique gene.
Answer 3: Thank you very much for this comment. It was clarified that BARF1 is a unique gene within the manuscript (line 161).
Comment 4: R should change to Rta.
Answer 4: Thank you very much for this comment. R was changed to Rta (Lines 160, 182 and 183).